# ReinboT: Amplifying Robot Visual-Language Manipulation with Reinforcement Learning

Hongyin Zhang [1,2]  Zifeng Zhuang [2]  Han Zhao [2]  Pengxiang Ding [2]  Hongchao Lu [2]  Donglin Wang [2]

## Abstract

Vision-Language-Action (VLA) models have shown great potential in general robotic decision-making tasks via imitation learning. However, the variable quality of training data often constrains the performance of these models. On the other hand, offline Reinforcement Learning (RL) excels at learning robust policy models from mixed-quality data. In this paper, we introduce **Rein**forced ro**bo**t GPT (**ReinboT**), a novel end-to-end VLA model that integrates the RL principle of maximizing cumulative reward. ReinboT achieves a deeper understanding of the data quality distribution by predicting dense returns that capture the nuances of manipulation tasks. The dense return prediction capability enables the robot to generate more robust decision-making actions, oriented towards maximizing future benefits. Extensive experiments show that ReinboT achieves state-of-the-art performance on the CALVIN mixed-quality dataset and exhibits superior few-shot learning and out-of-distribution generalization capabilities in real-world tasks.

## 1. Introduction

Research on vision-language-action (VLA) models for general embodied intelligence in robotics has recently flourished (Brohan et al., 2022; 2023). VLA models are usually based on the imitation learning paradigm, where a pre-trained vision-language model is post-trained on downstream robotic data (Ding et al., 2024; Zhao et al., 2025b). While semantic generalization has improved in VLA models through extensive robotic training data, a critical gap persists in their manipulation accuracy for downstream tasks (Brohan et al., 2023; Black et al.; Li et al., a).

---
[1]Zhejiang University, Hangzhou, China [2]Westlake University, Hangzhou, China. Correspondence to: Donglin Wang <wangdonglin@westlake.edu.cn>.

*Proceedings of the 42nd International Conference on Machine Learning*, Vancouver, Canada. PMLR 267, 2025. Copyright 2025 by the author(s).

An important reason that limits the performance of VLA models is that the quality of training data sources is usually uneven, even if they come from successful demonstrations (Hejna et al.). Although recent imitation learning methods can effectively replicate the distribution of demonstrations (Vuong et al., 2023; Brohan et al., 2023; Zhang et al., 2025), they have difficulty distinguishing between uneven data quality and making full use of mixed-quality data (Bai et al., 2025). On the other side, offline Reinforcement Learning (RL) algorithms aim to leverage previously collected data without the need for online data collection (Levine et al., 2020). Despite initial attempts to integrate VLA with RL (Mark et al., 2024; Zhai et al., 2024; Zhao et al., 2025a; Guo et al., 2025), the design of widely applicable dense rewards for visual-language manipulation tasks and the incorporation of the RL concept of maximizing benefits into the VLA model remain underexplored.

To this end, we propose **Rein**forced ro**bo**t GPT (**ReinboT**), a novel end-to-end VLA model to implement the RL concept of maximizing dense returns. Specifically, we efficiently and automatically decompose the long-horizon manipulation task trajectory into multiple trajectory segments containing only a single sub-goal, and design a dense reward that captures the characteristics of the manipulation task. In fact, complex robot manipulation tasks need to consider many factors, such as tracking targets, reducing energy consumption, and maintaining flexible and stable behavior. Therefore, the design principle of proposed reward densification method is based on these considerations and remains widely applicable to various manipulation tasks.

In terms of ReinboT algorithm design, we consider that accurate estimation of the value function in RL algorithms has always been a thorny problem, especially in the Transformer architecture (Parisotto et al., 2020; Davis et al., 2021). Therefore, we utilize cumulative rewards (i.e., ReturnToGo (Chen et al., 2021)) as a new modality data to characterize data quality characteristics based on the constructed dense reward. Inspired by previous work (Zhuang et al., 2024), we model the maximum return sequence over the joint distribution of language commands, image states (and proprioception), actions, and ReturnToGo. This is a supervised paradigm that integrates the RL goal of predict-

ing the maximum return within the distribution given the current conditions, and thereby considering the likelihood of maximizing actions. Specifically, we utilize expectile regression (Aigner et al., 1976; Sobotka & Kneib, 2012) to make the predicted return as close as possible to the maximum return that can be achieved under the current goals and states. With this ability, ReinboT can predict the maximum return during inference to guide the execution of better actions. Overall, the core contributions of this paper include:

- We propose ReinboT, a novel end-to-end VLA model that integrates RL returns maximization to enhance robotic manipulation capabilities.

- We introduce a reward densification method that enables ReinboT to gain deep insights into data quality for more robust learning.

- Extensive experiments demonstrate ReinboT's state-of-the-art performance, significantly outperforming baselines in both simulated and real-world tasks.

## 2. Related Work

### 2.1. Offline RL via Sequence Modeling

After the emergence of Transformer (Vaswani et al., 2017) as an efficient sequence modeling model, a large number of works (Chen et al., 2021; Yamagata et al., 2022; Janner et al., 2021; Zhuang et al., 2024; Shafiullah et al., 2022; Hu et al., 2024) have explored the application of sequence models as agent policies to RL decision tasks. The Decision Transformer (Chen et al., 2021) (DT) trains the context-conditional policy model on offline datasets through a supervised learning paradigm, conditioned on historical observations and ReturnToGo, and outputs the actions that the policy model should perform. Rein*for*mer (Zhuang et al., 2024) further introduces the concept of maximizing returns on the basis of DT. During training, Rein*for*mer not only predicts actions conditioned on the ReturnToGo in offline data, but also predicts the maximized ReturnToGo that the policy model may subsequently obtain under observation. Different from these task-specific studies, our work aims to implement the RL return maximization concept in a general-purpose VLA model to enhance the robot's long-horizon manipulation capabilities.

### 2.2. VLA Model Integrating with RL

Recent work has initially combined VLA with RL to study how to further improve the manipulation accuracy and adaptability of VLA models while still retaining their best advantages in scale and generalization. In these works, the source of the reward signal is either a sparse form of whether the goal is reached (Chebotar et al., 2023; Nakamoto et al.; Mark et al., 2024), or the number of steps to reach the goal (Yang et al.), or the distance to the goal is calculated with the help of LLM models and other pre-trained visual models (Zhang et al., 2024). However, these reward designs face the credit assignment problem that has not been fully solved in RL (Sutton, 1984), or are limited by the hallucination problem of LLM (Zhang et al., 2023).

In terms of combining with RL algorithms, these works mainly fine-tune existing VLA models that have undergone imitation learning, including introducing Q-functions to correct action distribution (Nakamoto et al.), screening out high-value action fine-tuning policies (Mark et al., 2024; Zhang et al., 2024), and fine-tuning according to human preferences (Chen et al., 2025). Moreover, a recent work (Chebotar et al., 2023) utilizes auto-regressive Q-functions to learn visual language manipulation, but the sequence length and inference time of their models increase significantly with the increase of action dimensions. Different from these studies, we aim to propose a new end-to-end reinforced VLA model based on dense rewards that capture the characteristics of manipulation tasks.

## 3. Preliminaries

### 3.1. Imitation Learning of VLA Model

As a typical VLA model for imitation learning, GR-1 (Wu et al.) demonstrates that visual robot manipulation can significantly benefit from large-scale video generation pre-training. Thanks to its flexible design, GR-1 can be seamlessly fine-tuned on robotic data after being pre-trained on large-scale video datasets. GR-1 is a GPT-style model that takes language instructions $l$, historical image observations $o_{t-h:t}$, and proprioception $s_{t-h:t}$ as input. It predicts robot actions and future images in an end-to-end manner $\langle \hat{o}_{t+1}, \hat{a}_t \rangle = \pi(l, \langle o, s \rangle_{t-h:t})$.

### 3.2. Max-Return Sequence Modeling

The sequence model DT (Chen et al., 2021) maximizes the likelihood of actions based on historical trajectories and ReturnToGo, which essentially transforms offline RL into supervised sequence modeling:

$$\mathcal{L}_a = \mathbb{E}_t \big[ -\log \pi_\theta(a_t | \langle s, g, a \rangle_{t-h:t-1}, s_t, g_t) \big], \quad (1)$$

where $g_t \doteq \sum_{j=t}^{T} r_j$ is ground-truth ReturnToGo in offline data. Rein*for*mer (Zhuang et al., 2024) integrates the goal of maximizing return into sequence models. Specifically, Rein*for*mer predicts the maximum returns that the current state might obtain within the data distribution represented by the dataset, rather than the ground-truth ReturnToGo of the current trajectory. Rein*for*mer achieves this implicitly through the minimizing of expectile regression loss:

$$\mathcal{L}_g = \mathbb{E}_t \big[ |m - \mathbb{1}(\Delta g < 0)|(\Delta g)^2 \big], \\ \text{with } \Delta g = g_t - \pi_\theta(\langle s, g, a \rangle_{t-h:t-1}, s_t), \quad (2)$$

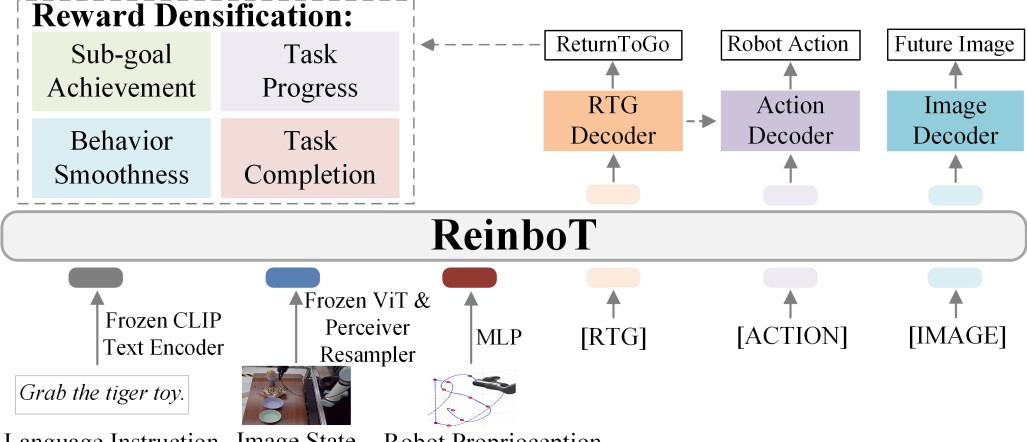

*Figure 1.* The proposed ReinboT model. We leverage CLIP (Radford et al., 2021) to encode robot language instructions, utilize ViT (Dosovitskiy et al., 2020; He et al., 2022) (and perceiver resampler (Jaegle et al., 2021)) to compress and encode the original pixel space of the image state, and utilize MLP to encode the robot proprioception. Moreover, based on the GPT-style transformer (Radford, 2018), we introduce three prediction token embeddings ([RTG], [ACTION] and [IMAGE]) to predict ReturnToGo, robot action, and future image state respectively. The last layer of hidden features in ReturnToGo decoder is further utilized to predict robot actions. The dense reward in ReturnToGo contains four aspects: sub-goal achievement, task progress, behavior smoothness and task completion.

where $\mathbb{1}(\cdot)$ is a binary indicator function, and $m \in (0, 1)$ is the hyperparameter of expectile regression. An excessively large parameter $m$ may cause the model to over-optimistically estimate the maximum possible ReturnToGo in the training data distribution. This will cause the prediction of a ReturnToGo outside the distribution during inference, which will negatively affect action generation. Rein*for*mer was trained by minimizing the sum of two loss functions $\mathcal{L} = \mathcal{L}_a + \mathcal{L}_g$. Compared with DT, one advantage of the Rein*for*mer is that it does not need to specify the initial value of ReturnToGo and the reward returned by the environment during inference. The Rein*for*mer can autoregressively predict the maximum ReturnToGo and action of the next step through two model inferences:

$$\begin{cases} \hat{g}_t = \pi(\langle s, g, a \rangle_{t-h:t-1}, s_t), \\ \hat{a}_t = \pi(\langle s, g, a \rangle_{t-h:t-1}, s_t, \hat{g}_t). \end{cases} \quad (3)$$

## 4. Methodology

In this paper, we aim to build a novel end-to-end VLA model that incorporates the principle of maximizing dense returns into robot visuomotor control, as shown in Fig. 1. First, we consider four main factors when designing dense rewards to capture the nature of the robot's long-horizon manipulation task (Sec. 4.1). Then in Sec. 4.2, we elaborate on how to build a novel end-to-end reinforced VLA model and test execution pipeline. Finally, we discuss and analyze how the proposed ReinboT organically integrates the principle of RL maximizing return (Sec. 4.3).

### 4.1. Reward Densification

For long-horizon visual-language manipulation tasks, VLA models are usually required to maintain robust and stable behavior at minimal energy cost while following the goal. Therefore, we mainly design a widely applicable dense reward around this principle to capture the nature of the manipulation task. Intuitively, in the robot trajectory, the reward that minimizes the state distance is a simple and effective scheme that encourages the robot to move directly to the target state. However, this reward is limited to the case when the task contains only one goal. For long-horizon tasks that require manipulating objects with multiple sub-goals, this reward will guide the robot to move directly to the final target state, resulting in failure (Zhao et al., 2024).

Therefore, we first adopt a heuristic method (James & Davison, 2022; Shridhar et al., 2023) to divide the long-horizon manipulation task into multiple sub-goal sequences and design a dense reward for each sequence. The heuristic process iterates over the states in each demonstrated trajectory and determines whether the state should be considered a critical state. The judgment is based on two main constraints: joint velocities close to zero and changes in gripper state. Intuitively, this occurs when the robot reaches a pre-grasp pose or transitions to a new task phase, or when grasping or releasing an object. Therefore, utilizing the critical state as a sub-goal is a natural and reasonable choice.

**Sub-goal achievement.** Both the image state $o_t$ and proprioception $s_t$ contain rich environmental perception information. Therefore, the sub-goal achievement reward $r_1$ covers

proprioceptive tracking, pixel intensity, image visual quality, and image feature points:

$$r_1 = e^{f_{\text{MSE}}(s_t, s_t^*)} + e^{f_{\text{MSE}}(o_t, o_t^*)}$$
$$+ e^{f_{\text{SSIM}}(o_t, o_t^*)} + e^{f_{\text{ORB}}(o_t, o_t^*)}. \quad (4)$$

We utilize Mean Square Error (MSE) to calculate the direct difference between the image state $o_t$ (and proprioception $s_t$) and the sub-goal image state $o_t^*$ (and sub-goal proprioception $s_t^*$), and utilize the Structural Similarity Index (SSIM) to measure the visual quality of the image. The Oriented FAST and Rotated BRIEF (ORB) (Rublee et al., 2011) algorithm utilized to calculate the reward focuses on the extraction and matching of image feature points. Specifically, we first detect the key points on the current image state and the sub-goal image state, perform feature matching and matching point screening, and finally calculate the similarity by the number of matching points.

**Task progress.** Considering that the impact of being divided into several sub-goal sequences on the overall trajectory is different. The later sequences are closer to the final target state. To reflect this, task progress reward $r_2$ is designed:

$$r_2 = \frac{n(s_t)}{|\{s^*\}|}, \quad (5)$$

where $n(s_t) = \{1, 2, \cdots, |\{s^*\}|\}$ indicates which sub-goal sequence the state $s_t$ is in. The closer the sub-goal sequence is to the final goal state, the greater the task progress reward.

**Behavior smoothness.** To promote a smooth and natural movement trajectory, we mainly consider suppressing the joint velocity $\dot{\mathbf{q}}$ and acceleration $\ddot{\mathbf{q}}$ of the robot arm movement and the rate of change of the action $\mathbf{a}_t$, thus punishing the trajectory movement that is too violent and stiff. The behavior smoothness reward $r_3$ is:

$$r_3 = -|\dot{\mathbf{q}}|^2 - |\ddot{\mathbf{q}}|^2 - |\mathbf{a}_{t-1} - \mathbf{a}_t|^2 - |\mathbf{a}_{t-2} - 2\mathbf{a}_{t-1} + \mathbf{a}_t|^2. \quad (6)$$

**Task completion.** For the visual language manipulation task, language instruction is regarded as a goal that matches the robot's behavior. The task completion reward $r_4$ is:

$$r_4 = \mathbb{1}\{\tau \text{ is successful}\}. \quad (7)$$

Here, $\mathbb{1}(\cdot)$ is a binary indicator function used to indicate whether a trajectory $\tau$ completes the instruction.

Based on these four main factors, the general dense reward captures the nature of the long-horizon visual-language manipulation tasks is:

$$r = \sum_{i=1}^{4} w_i r_i, \quad (8)$$

where $\{w_i\}_{i=1}^{4}$ is the reward weight, which can ensure that each reward component maintains a comparable order of

magnitude between demonstration trajectories. By utilizing the designed reward signal, ReinboT can have a broader and deeper understanding and recognition of the quality distribution of training data, thereby guiding the robot to perform more robust and stable robot decision actions.

### 4.2. End-to-end Reinforced VLA model

Through the proposed dense reward, we can obtain ReturnToGo (RTG) $g_t = \sum_{j=t}^{T} r_j$ for long-horizon visual-language manipulation tasks. We further explain how to build a novel end-to-end reinforced VLA model to implement the RL principle of maximizing return. The proposed ReinboT model utilizes GPT-style transformer (Radford, 2018) as the backbone network $\pi_\theta$ because it can flexibly and efficiently use different types of modal data as input and output. The CLIP (Radford et al., 2021) is utilized to encode language instructions, ViT (Dosovitskiy et al., 2020; He et al., 2022) (and perceiver resampler (Jaegle et al., 2021)) is utilized to compress and encode image states, and MLP is utilized to encode proprioception. We introduce action and image token embeddings ([ACTION] and [IMAGE]) and predict robot actions and future image states through an action decoder $P_\omega$ and an image decoder $P_\nu$, respectively. Most importantly, we treat ReturnToGo as a novel modality of data and learn ReturnToGo prediction token embedding [RTG]. We predict the maximized return given the language instruction $l$, image state $o$, and proprioception $s$ through the ReturnToGo decoder $P_\varphi$:

$$\mathcal{L}_{\text{RTG}} = \mathbb{E}_t \left[ |m - \mathbb{1}(\Delta g < 0)|(\Delta g)^2 \right],$$
$$\text{with } \Delta g = g_t - P_\varphi \left[ \pi_\theta (\langle s, o \rangle_{t-h+1:t}, l) \right]. \quad (9)$$

The loss function $\mathcal{L}$ of the ReinboT model comprises the ReturnToGo loss $\mathcal{L}_{\text{RTG}}$, the arm action loss $\mathcal{L}_{\text{arm}}$, the gripper action loss $\mathcal{L}_{\text{gripper}}$, and the future image loss $\mathcal{L}_{\text{image}}$:

$$\mathcal{L} = \lambda \mathcal{L}_{\text{RTG}} + \mathcal{L}_{\text{arm}} + 0.01 \mathcal{L}_{\text{gripper}} + 0.1 \mathcal{L}_{\text{image}}, \quad (10)$$

where $\lambda$ is the ReturnToGo loss weight, and the loss weights for other modalities follow previous work (Li et al., 2025). $\mathcal{L}_{\text{arm}}$ is a smooth-$\mathcal{L}_1$ loss, $\mathcal{L}_{\text{gripper}}$ is a cross entropy loss, and $\mathcal{L}_{\text{image}}$ is a pixel-level MSE.

When designing how to predict action $a$ with feature information containing ReturnToGo, we make modular designs in the ReinboT network structure. Specifically, we first input the language instruction $l$, image state $o_{t-u+1:t}$ and proprioception $s_{t-u+1:t}$ into the backbone network $\pi_\phi$, and obtain the features $h_{t:t+k-1}^{\text{RTG}}$ and $h_{t:t+k-1}^{\text{action}}$ corresponding to [RTG] and [ACTION] token embeddings:

$$h_{t:t+k-1}^{\text{RTG}}, h_{t:t+k-1}^{\text{action}} = \pi_\phi(l, o_{t-u+1:t}, s_{t-u+1:t}). \quad (11)$$

The feature $h_{t:t+k-1}^{\text{RTG}}$ is then input into the ReturnToGo decoder $P_\varphi$ to obtain the last layer of hidden features $\hat{g}_{t:t+k-1}^{\text{hidden}}$:

$$\hat{g}_{t:t+k-1}^{\text{hidden}} = P_\varphi(h_{t:t+k-1}^{\text{RTG}}). \quad (12)$$

The hidden features $\hat{g}_{t:t+k-1}^{\text{hidden}}$ is concatenated with the action features $h_{t:t+k-1}^{\text{action}}$ and are further input into the action decoder $P_\omega$ to predict the action $\hat{a}_{t:t+k-1}$:

$$\hat{a}_{t:t+k-1} = P_\omega(h_{t:t+k-1}, \hat{g}_{t:t+k-1}^{\text{hidden}}). \quad (13)$$

The modular design in ReinboT allows us to obtain robot actions with only single model inference, thus enjoying higher inference efficiency than the Rein*for*mer model. The more benefit of this design is that during the inference phase, we do not need to manually set the initial value of ReturnToGo like the DT model. This is crucial for actual deployment because it greatly alleviates the tediousness of manual parameter adjustment, and the actual deployment environment cannot directly obtain rewards to a large extent. The ReinboT inference pipeline has been summarized in Alg. 1. The implementation details are in Appendix Sec. A.1.

---

**Algorithm 1** ReinboT: Test-time Execution
---
1: **ReinboT model** $\pi_\phi$, $P_\varphi$, $P_\omega$, initial image state $o_{0,\text{test}}$, initial proprioception $s_{0,\text{test}}$, language instruction $l_{\text{test}}$, and environment Env. // ReturnToGo initialization value is not required.
2: $t \leftarrow 0$
3: **while** $t \leq T_{\text{test}}$ **do**
4:     Calculate ReturnToGo and action features by Eq. 11
5:     Calculate ReturnToGo last layer of hidden features by Eq. 12
6:     Calculate robot action $\hat{a}_{t:t+k-1}$ by Eq. 13
7:     Interact with Env $o_{t+1,\text{test}}, s_{t+1,\text{test}} \leftarrow \text{Env.Step}(\hat{a}_t)$ // Reward is not required.
8:     $t \leftarrow t + 1$
9: **end while**
---

### 4.3. Discussion and Analysis of ReinboT

Compared with common end-to-end VLA models, the most significant feature of proposed ReinboT is the additional introduction of ReturnToGo loss (Eq. 9), and the action is affected by the ReturnToGo token $\hat{g}_{t:t+k-1}^{\text{hidden}}$ (Eq. 13). We will subsequently analyze how this framework achieves RL return maximization, as well as the differences and advantages compared to return maximization in classic RL.

Take a date point $\left(l, \langle o, s\rangle_{t-h+1}, g_t, a_t\right)$ for example. The loss of ReturnToGo is implemented based on expectile regression, with a key parameter $m$. When $m = 0.5$, the expectile regression degenerates into MSE, and the predicted ReturnToGo token $\hat{g}_t$ approaches the ground-truth $g_t$. At this point, ReinboT degenerates into the paradigm of imitation learning, also the common end-to-end VLA models equipped with ReturnToGo token prediction. When $m > 0.5$, the expectile regression will predict the $\hat{g}_t$ greater than $g_t$, which is called return maximization. This maximized return guides the ReinboT to predict a better action.

However, blindly increasing $m$ will lead to the model's overly optimistic estimate of the maximum return that can be achieved in the training data distribution. Related theoretical analysis is in Rein*for*mer (Zhuang et al., 2024).

In the classic RL algorithm, maximizing the Q-value is utilized to achieve the best policy model. This implies that applying RL in VLA necessitates the introduction of an additional RL loss function. Such an addition may pose obstacles to the learning process of models like transformers (Mishra et al., 2018; Parisotto et al., 2020; Davis et al., 2021). In contrast, our return condition maximization circumvents the need to incorporate the RL-specific loss.

## 5. Experiments

In this section, we explore how the proposed ReinboT model can effectively implement the RL principle of maximizing return to enhance robotic vision-language manipulation tasks. To this end, our experiments aim to investigate the following questions: **1)** Does ReinboT show better generalization ability and higher success rate when performing long-horizon tasks compared to baseline algorithms? **2)** How important is the dense reward component to the overall generalization performance of ReinboT? **3)** What are the characteristics of maximizing return prediction in ReinboT? **4)** Can ReinboT complete few-shot learning and out-of-distribution (OOD) generalization in real-world scenarios?

### 5.1. Generalization Evaluation on Mixed-quality Data

**Setting.** We first construct a mixed-quality dataset based on CALVIN (Mees et al., 2022), which contains long-horizon manipulation tasks, to examine the performance of the proposed ReinboT and baseline algorithms. This dataset contains a small amount of data with language instructions in CALVIN ABC (about 50 trajectories per task) and a large amount of autonomous data without language instructions. In addition to the original data collected by human teleoperation without language instructions in CALVIN (more than 20,000 trajectories), the autonomous data also contains failure data generated by the interaction between the trained VLA behavioral policy RoboFlamingo (Li et al., b) and the environment CALVIN D (more than 10,000 trajectories). To promote data diversity, different degrees of Gaussian noise (0.05, 0.1, and 0.15) are added to the actions of the RoboFlamingo policy model during the interaction. We study training on this mixed-quality data, then fine-tune a small amount of data with language instructions, and finally test the generalization performance on CALVIN D. The subgoal division and dense reward examples of mixed-quality training data are in Appendix Fig. 8 ∼ 11. Tab. 1 shows the success rate of each language instruction in the chain and the Average Length (AL) of the completed tasks.

*Table 1.* Generalization performance comparison of models trained on CALVIN mixed-quality data to test environment D.

| Algorithms | No. of Instructions Chained | | | | | Avg. Length (↑) |
|---|---|---|---|---|---|---|
| | 1 | 2 | 3 | 4 | 5 | |
| RoboFlamingo (annotated data) | 0.55 | 0.19 | 0.07 | 0.02 | 0.00 | 0.83 |
| GR-1 (annotated data) | 0.67 | 0.37 | 0.20 | 0.11 | 0.07 | 1.41 |
| PIDM (annotated data) | 0.60 | 0.45 | 0.32 | 0.23 | 0.13 | 1.73 |
| GR-1 | 0.62 | 0.31 | 0.18 | 0.14 | 0.10 | 1.36 |
| GR-MG | 0.65 | 0.35 | 0.24 | 0.11 | 0.05 | 1.41 |
| RWR (sparse) | 0.63 | 0.36 | 0.21 | 0.12 | 0.07 | 1.38 |
| RWR (sub-goal, sparse) | 0.71 | 0.46 | 0.27 | 0.19 | 0.11 | 1.73 |
| RWR (dense, single) | 0.75 | 0.52 | 0.27 | 0.18 | 0.11 | 1.82 |
| ReinboT (sparse) | 0.70 | 0.44 | 0.29 | 0.19 | 0.12 | 1.74 |
| ReinboT (sub-goal, sparse) | 0.74 | 0.50 | 0.28 | 0.17 | 0.12 | 1.80 |
| ReinboT (dense, single) | 0.77 | 0.53 | 0.32 | 0.18 | 0.11 | 1.90 |
| ReinboT (dense, full) | **0.79** | **0.58** | **0.40** | **0.28** | **0.21** | **2.26** |

*Table 2.* Ablation experiments are conducted to verify the necessity of the designed reward components.

| | No. of Instructions Chained | | | | | Avg. Length (↑) |
|---|---|---|---|---|---|---|
| | 1 | 2 | 3 | 4 | 5 | |
| ReinboT (dense, full) | **0.79** | **0.58** | **0.40** | **0.28** | **0.21** | **2.26** |
| W/o ReturnToGo | 0.65 | 0.36 | 0.19 | 0.11 | 0.06 | 1.36 (-39.8%) |
| W/o sub-goal achievement $r_1$ | 0.72 | 0.50 | 0.32 | 0.20 | 0.12 | 1.87 (-17.2%) |
| W/o task progress $r_2$ | 0.75 | 0.48 | 0.29 | 0.17 | 0.10 | 1.79 (-20.8%) |
| W/o behavior smoothness $r_3$ | 0.73 | 0.50 | 0.32 | 0.21 | 0.14 | 1.90 (-15.9%) |
| W/o task completion $r_4$ | 0.75 | 0.50 | 0.33 | 0.21 | 0.14 | 1.93 (-14.6%) |

**Baselines.** To thoroughly evaluate the effectiveness of the proposed ReinboT model, we consider some representative baseline algorithms and reward design methods, including RoboFlamingo (Li et al., b), GR-1 (Wu et al.), PIDM (Tian et al., 2024) (three imitation learning types), GR-MG (Li et al., 2025) (hierarchical imitation learning type), and RWR (Peters & Schaal, 2007) (offline RL type). "annotated data" means that the model is trained only on a small amount of data with text annotations (about 50 trajectories per task). "sparse" means utilizing sparse rewards, that is, the reward of the last three steps of a successful trajectory is 1, and the rest is 0 (Nakamoto et al.). "sub-goal, sparse" means utilizing sparse rewards, that is, the reward of the last three steps of a successful trajectory and the sub-goal state is 1, and the rest is 0. "dense, single" means utilizing the dense reward we proposed, and the final calculated single-dimensional scalar return is utilized when calculating the ReturnToGo loss. "dense, full" means utilizing the dense reward we proposed, and the predicted ReturnToGo is a vector containing the calculated single-dimensional scalar return and each return component. The baseline details are introduced in Appendix Sec. A.2.

**Generalization performance comparison.** Tab. 1 shows that among the models trained only on data with text annotations, PIDM integrates vision and action into a closed

loop and achieves better generalization performance than the baselines RoboFlamingo and GR-1. However, limited to the imitation learning type, the performance of PIDM (AL is 1.73) is lower than the offline RL type model RWR and ReinboT (AL is 1.82 and 2.26, respectively). Moreover, the performance of GR-1 (AL is 1.36), which can obtain all the training data information, is slightly lower than the performance of GR-1 trained only on training data with text annotations (AL is 1.41). GR-MG is also limited by the low-level policy based on imitation learning. Therefore, the VLA model under the imitation learning paradigm only performs maximum likelihood on the original training data distribution, which is difficult to capture and fully utilize the characteristics of the mixed quality distribution, resulting in unsatisfactory performance.

For ReinboT and RWR, our dense reward improves performance better than sparse rewards. Predicting each component of ReturnToGo can further improve the generalization ability of ReinboT (AL increased from 1.90 to 2.26). Therefore, the proposed reward has a deeper and more detailed representation of the data quality distribution, thus bringing denser supervision signals to the training of the VLA model. The ReinboT can effectively implement the idea of RL to leverage dense return maximization to enhance long-horizon visual-language manipulation tasks.

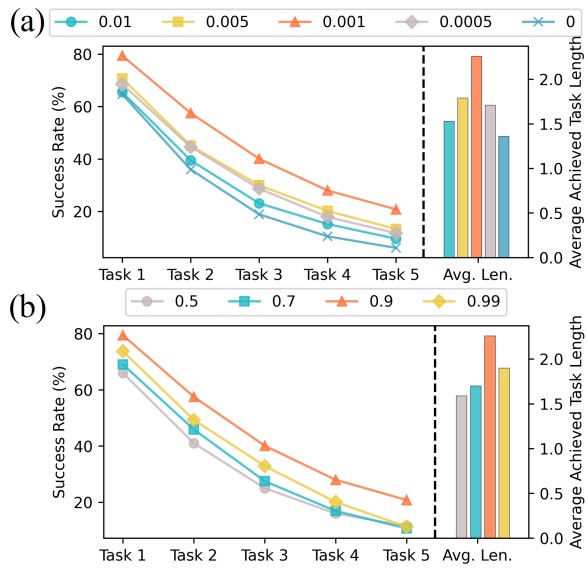

*Figure 2.* (a) Impact of different values of ReturnToGo $\mathcal{L}_{\text{RTG}}$ loss weight $\lambda$. (b) Impact of different values of the expectile regression parameter $m$ in the ReturnToGo $\mathcal{L}_{\text{RTG}}$ loss function.

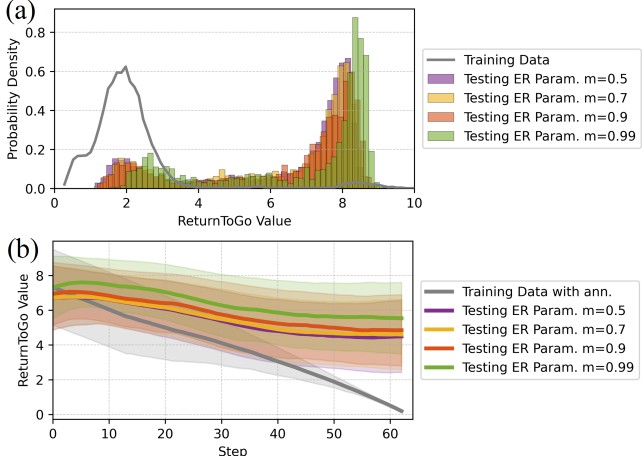

*Figure 3.* (a) Distribution of ground-truth ReturnToGo of CALVIN mixed-quality training data and distribution of the maximized ReturnToGo predicted by the ReinboT when interacting with the test environment D. (b) Comparison of ReturnToGo in the training data with text annotations in mixed-quality data and the maximized ReturnToGo predicted by the ReinboT at the interaction time step. The impact of different values of the Expectile Regression (ER) parameter $m$ in the $\mathcal{L}_{\text{RTG}}$ loss function is investigated.

### 5.2. Ablation Study

**Ablation of dense reward component.** We conduct ablation experiments on ReinboT trained on CALVIN mixed-quality data and tested on environment D to evaluate the contribution of each reward component to the model generalization (Tab. 2). If there is no ReturnToGo modal information in ReinboT, the metric AL will drop sharply by 39.8%. If there is a lack of reward component in ReinboT, its performance will be lost to varying degrees. The most significant impact is on task progress $r_2$ (reduced by 20.8%), followed by sub-goal achievement $r_1$ (reduced by 17.2%). The reward components behavior smoothness $r_3$ and task completion $r_4$ have similar impacts, decreasing by 15.9% and 14.6% respectively. Therefore, each reward component can help the model to deeply identify various aspects of data quality and has a significant impact on the robot's generalization performance.

**Performance impact of hyperparameters $\lambda$ and $m$.** We further conduct ablation experiments on $\lambda$ and $m$ introduced in ReinboT, trained on CALVIN mixed-quality data and tested on environment D, to explore their impact on model performance (Fig. 2). The hyperparameter $\lambda$ is utilized to make a trade-off between the model's prediction of ReturnToGo and other modalities. The expectile regression parameter $m$ is utilized to control the model's sensitivity to different expectation levels, thereby adjusting the model's fitting characteristics for the ReturnToGo distribution. Experimental results show that when $\lambda = 0.001$ and $m = 0.9$, ReinboT achieves the best performance, which is the default setting for our model unless otherwise specified.

**Properties of the predicted maximized RL return.** To analyze the underlying reasons for the performance improvement of the proposed RinboGPT model, we explore the properties of the predicted maximized RL return (Fig. 3). The sample size of the training data is in Appendix Fig. 7. The results show that as the expectile regression parameter $m$ increases, the ReturnToGo distribution shifts towards a larger value. Therefore, ReinboT can effectively identify and distinguish the quality distribution of training data and predict the robot action that maximizes return in the current (and historical) state as much as possible. This means that when the robot performs an action, it will consider maximizing the long-horizon benefits in the future period, rather than only considering the current (and historical) state in the short term. This capability can effectively enhance the generalization performance of the ReinboT model in long-horizon manipulation tasks.

Moreover, we can find that the model performance with parameter $m = 0.99$ is lower than that with $m = 0.9$ (Fig. 2(b)), although the former has a higher ReturnToGo tendency (Fig. 3). This indicates that too large $m$ will cause ReinboT to overestimate the maximum ReturnToGo that can be achieved in the training data distribution. The ReinboT has difficulty responding to the predicted over-estimated ReturnToGo, resulting in performance degradation.

Few-shot Learning      Out-of-distribution Generalization

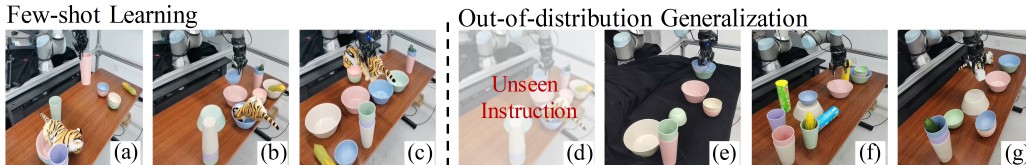

*Figure 4.* Few-shot learning and OOD generalization evaluation scenarios for real-world tasks. Few-shot learning evaluation scenarios include cup grasping (a), bowl grasping and placing (b), and plush toy grasping and placing (c). OOD generalization evaluation scenarios include unseen language instructions (d), desktop backgrounds (e), distractors (f), and manipulated objects (g).

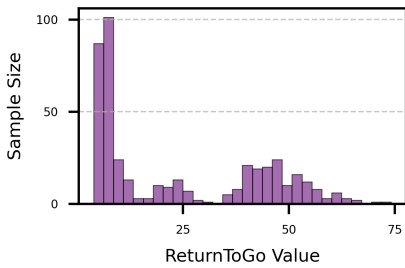

*Figure 5.* Distribution of successful realistic trajectories.

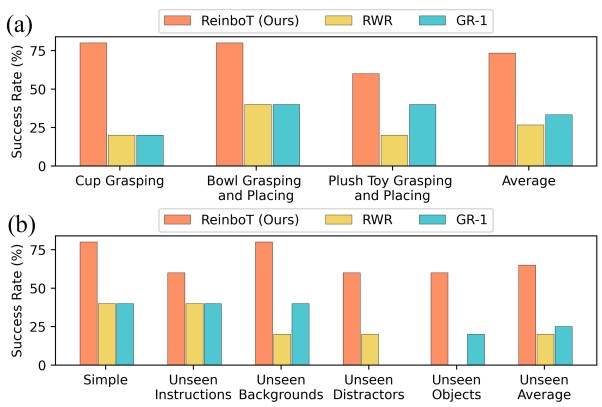

*Figure 6.* (a) Comparison of few-shot learning performance. (b) Generalization comparison on simple and unseen tasks.

## 5.3. Evaluation on Real-world Tasks

**Settings.** We conduct evaluations on real-world tasks to examine whether the proposed ReinboT can perform effective few-shot learning and generalization in realistic scenarios. Specifically, we consider the picking and placing tasks of objects such as cups, bowls, and stuffed toys on a robotic arm UR5. The total number of successful trajectories collected is approximately 530 (data distribution is in Fig. 5), and the model is first trained on these data. The sub-goal division and dense reward examples of successful training data on real-world UR5 are in Appendix Fig. 12 ∼ 15. For few-shot learning evaluation, we consider three object grasping and placing tasks (Fig. 4(a-c)). Each task contains only 30 successful trajectories, and the model is fine-tuned on these three tasks. For OOD generalization evaluation, we consider scenes with unseen instructions, backgrounds, distractors and manipulated objects (Fig. 4(d-g)).

**ReturnToGo distribution of successful trajectories.** Fig. 5 shows the ReturnToGo distribution of successful trajectories in reality. The result shows that even if the training data are all successful trajectories, their quality distribution is still uneven under the dense reward metric we proposed. Therefore, it is necessary to introduce RL ideas into the VLA model to deeply identify the data distribution and guide the prediction of actions that maximize data quality.

**Real machine comparison.** The quantitative performance comparison of real-world tasks is in Fig. 6, and the qualitative results are in Appendix Fig. 16. The experimental results show that the proposed ReinboT has excellent few-

shot learning and OOD generalization performance in realistic scenarios, and significantly outperforms the baseline methods. This is due to ReinboT's ability to effectively consider maximizing future returns. RWR performs on par with the GR-1. This may be due to the overfitting of RWR to the training data and its reliance on reweighting the data, which may lead to optimization problems when the data distribution is uneven or the amount of data is insufficient.

## 6. Conclusion

This paper internalizes the principle of maximizing return in RL into the VLA framework, thereby enhancing the robot's long-horizon manipulation capabilities. The proposed ReinboT can predict the maximum dense return that depicts important information of the manipulation task, thus having a deep and detailed understanding of data quality. This ability allows the robot to consider not only the current (and historical) state, but also the future dense benefits when taking decision actions. Compared with baselines, ReinboT achieves excellent performance in both simulated and real-world visual-language manipulation tasks. Our work advances robot visual-language manipulation capabilities, contributing to embodied general intelligence. A promising work is to consider the scaling of models and data to cope with the rich and diverse robotic tasks in the real world.

## Acknowledgments

This work was supported by the National Science and Technology Innovation 2030 - Major Project (Grant No. 2022ZD0208800), and NSFC General Program (Grant No. 62176215).

## Impact Statement

This research introduces Reinforced robot GPT (ReinboT), a novel Vision-Language-Action model that integrates reinforcement learning principles to address the limitations of variable training data quality in robotic decision-making tasks. By predicting dense returns that capture the nuances of manipulation tasks, ReinboT achieves a deeper understanding of data quality distribution and generates more robust decision-making actions aimed at maximizing future benefits. The model demonstrates state-of-the-art performance on the CALVIN mixed-quality dataset and superior few-shot learning and out-of-distribution generalization capabilities in real-world tasks. This advancement has the potential to significantly enhance the adaptability and efficiency of robotic systems in complex, real-world environments, paving the way for more reliable and versatile robotic applications across various industries.

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

# A. Appendix

## A.1. Implementation Details

**Network structure and training process**. The network backbone of ReinboT adopts the GPT2 (Radford et al., 2019) structure, and the ReturnToGo decoder and image decoder adopt the Transformer structure. In terms of the action decoder of ReinboT, we follow previous work (Zhao et al.) and predict the robot action trajectory through Conditional Variational AutoEncoder (CVAE) (Sohn et al., 2015; Kingma, 2013). Specifically, the cVAE encoder is utilized to encode the action trajectory into a style vector embedding. The style vector embedding, the output embedding of the [ACTION] token, and the $k$ learnable token embeddings are concatenated together. Then we input them into the Transformer to predict the $k$-step action trajectory. The network design hyperparameters of ReinboT are shown in Tab. 3.

*Table 3.* Network hyperparameters configuration.

| Parameter | Value |
|---|---|
| Number of action prediction horizon | 5 for CALVIN; 64 for UR5 |
| Number of image (proprioception) history stack | 10 |
| Latent dimension of action encoder | 32 |
| Hidden layer dimension of action encoder/decoder | 128 |
| Hidden layer dimension of RTG decoder | 128 |
| Visual feature dimension | 768 |
| Language feature dimension | 768 |
| Embedding dimension | 384 |
| Number of layers in backbone | 12 |
| Number of attention heads | 12 |
| Activation function | ReLU |

To train the ReinboT model more efficiently, we initialize its weights with the pre-trained model weights, which are derived from the generated video pre-training on the Ego4d (Grauman et al., 2022) dataset consistent with GR-1. The ReinboT model allows the use of training data without language instructions. In the specific implementation, we provide an empty string as the language instruction input of the model. The training hyperparameters are shown in Tab. 4.

*Table 4.* Training hyperparameters configuration.

| Parameter | Value |
|---|---|
| ReturnToGo loss weight $\lambda$ | 0.001 |
| Expectile regression parameter $m$ | 0.9 |
| Gradient clip | 1.0 |
| Epochs | 50 |
| Warm-up epochs | 1 |
| Batch size | 32 |
| Learning rate | 0.001 |
| Weight decay | 0.01 |
| Dropout rate | 0.1 |
| Reward weight $w_{i=1}^{4}$ | $0.1, 0.1, 0.01, 0.1$ |
| Optimizer | Adam ($\beta_1 = 0.9, \beta_2 = 0.999$) (Kingma, 2014) |

## A.2. Baselines Introduction

To evaluate the effectiveness of the proposed ReinboT model, some representative baseline algorithms and reward design methods are considered: **1) RoboFlamingo**: RoboFlamingo (Li et al., b) is a VLA model that leverages pre-trained VLMs for single-step visual-language understanding and models sequence history information with an explicit policy head. **2) GR-1**: GR-1 (Wu et al.) is a simple and effective imitation learning method that utilizes a pre-trained video model to enhance action generation. **3) GR-MG**: GR-MG (Li et al., 2025) leverages partially-annotated data by conditioning on text instructions and goal images, using a diffusion model to generate goal images during inference. **4) PIDM**: PIDM (Tian et al.,

2024) presents an end-to-end paradigm that utilizes inverse dynamics models conditioned on the robot's forecasted visual states to predict actions, integrating vision and action in a closed loop for scalable robotic manipulation. **5) RWR**: The RWR (Peters & Schaal, 2007) we reproduced is an offline RL algorithm that aims to directly optimize the VLA policy model by maximizing the cumulative reward while bypassing the thorny RL value function estimation problem. The discounting factor $\gamma$ used in our experiment is set as $0.9$. The update gradient for the VLA tasks we reproduced when calculating the action loss function $\mathcal{L}_a$ is:

$$\nabla_\pi \mathcal{L}_a = \frac{1}{N} \sum_\tau \nabla_\pi \log \pi(a|l, \langle o, s \rangle_{t-h:t})[\sum_{i=t}^{T} \gamma^{i-t} \cdot r(l, \langle o, s, a \rangle_{t-h:t})]. \tag{14}$$

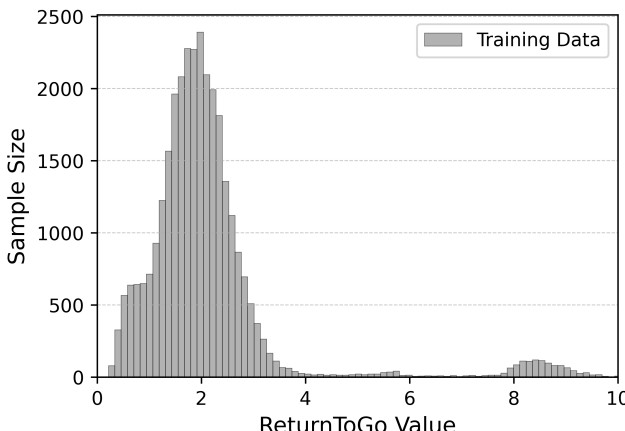

*Figure 7.* Distribution of ground-truth ReturnToGo of CALVIN mixed-quality training data.

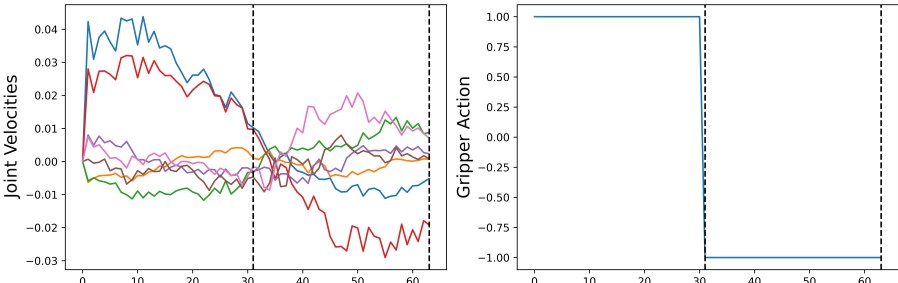

*Figure 8.* The sub-goal division of long-horizon tasks with language instructions of "*slide the door to the left*" in CALVIN mixed-quality training data. The black vertical dashed line represents the sub-goal of the trajectory (the same below).

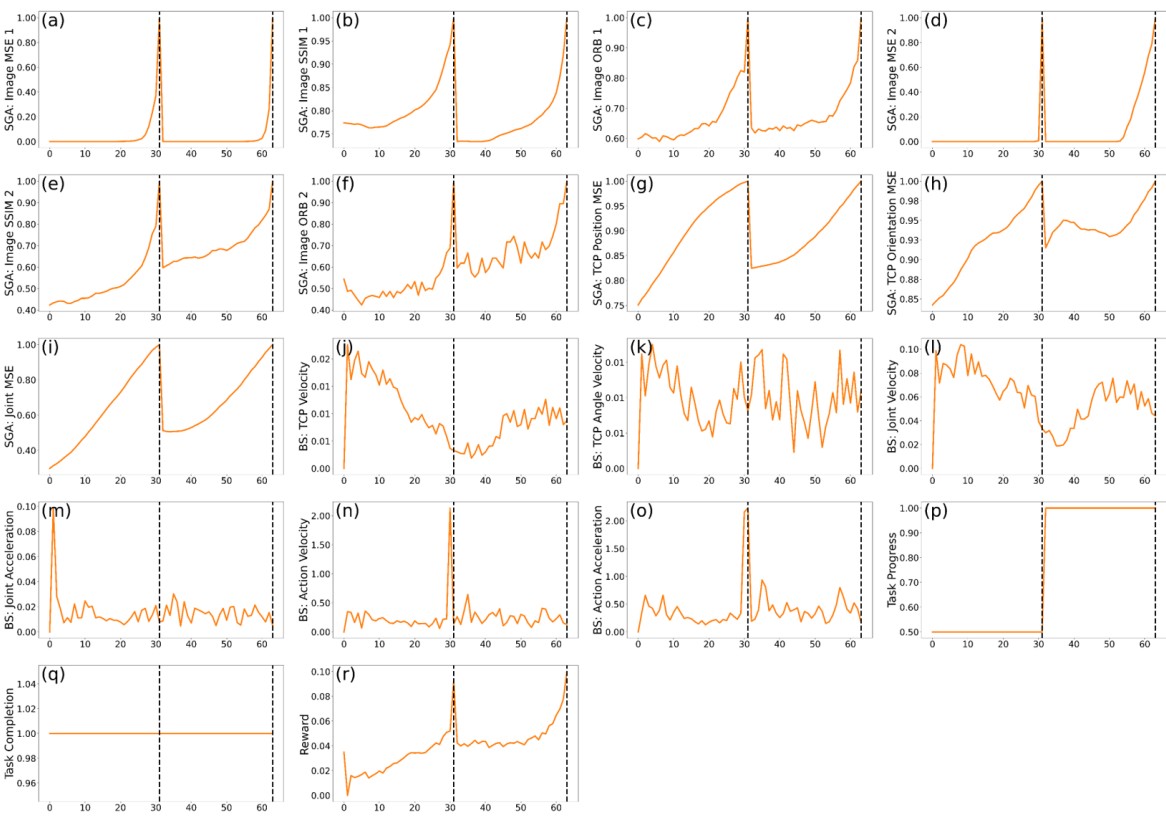

*Figure 9.* The dense reward and reward component of long-horizon tasks with language instructions of "*slide the door to the left*" in CALVIN mixed-quality training data. SGA: Sub-goal Achievement. BS: Behavior Smoothness. Same below.

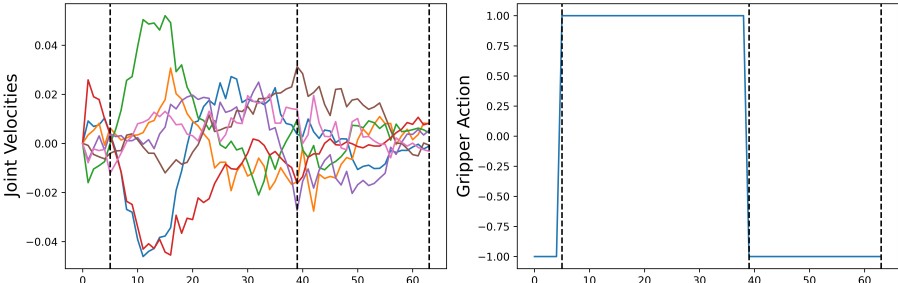

*Figure 10.* The sub-goal division of long-horizon tasks with language instructions of "*turn on the yellow lamp*" in CALVIN mixed-quality training data. The black vertical dashed line represents the sub-goal of the trajectory (the same below).

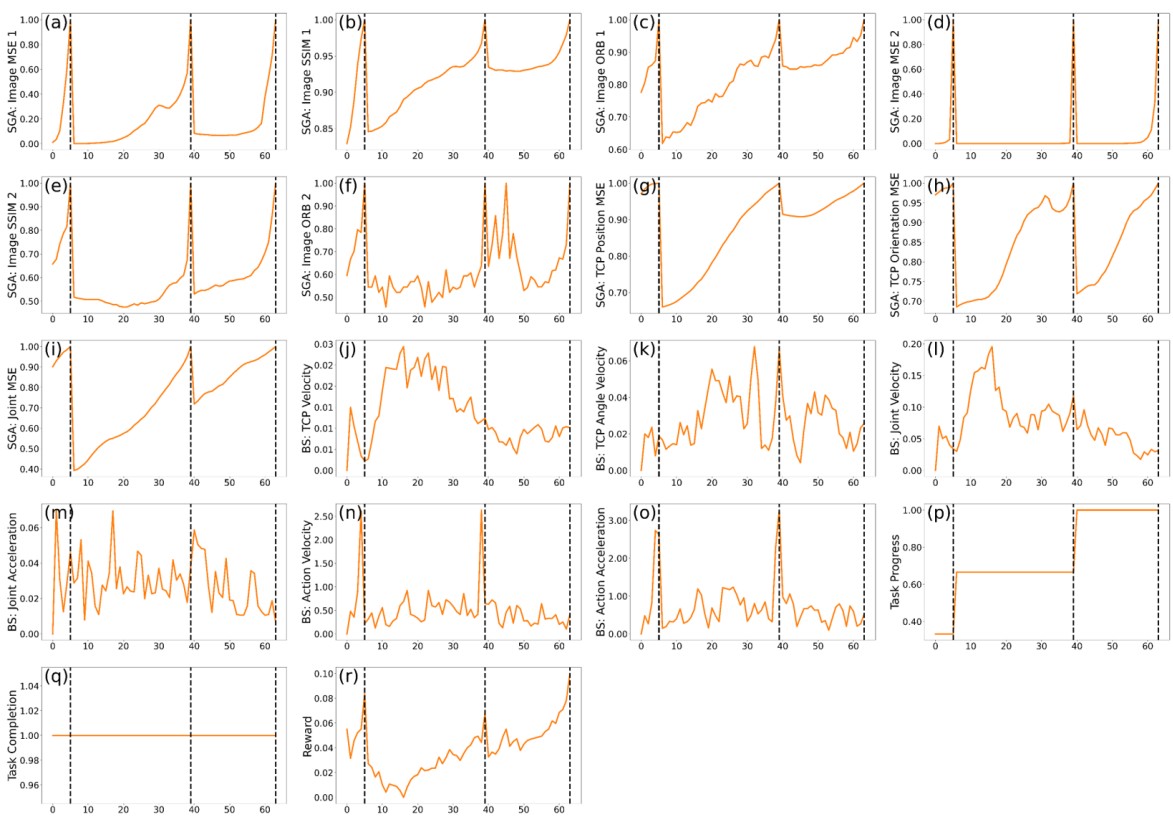

*Figure 11.* The dense reward and reward component of long-horizon tasks with language instructions of "*turn on the yellow lamp*" in CALVIN mixed-quality training data.

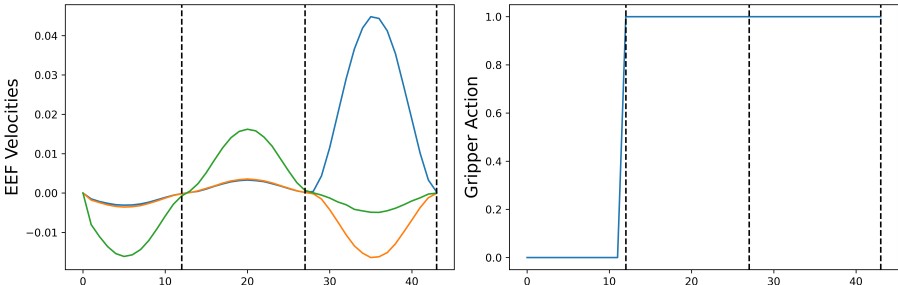

*Figure 12.* The sub-goal division of long-horizon tasks with language instructions of "*Pick up the green cup for me*" in the real-world UR5 successful training data.

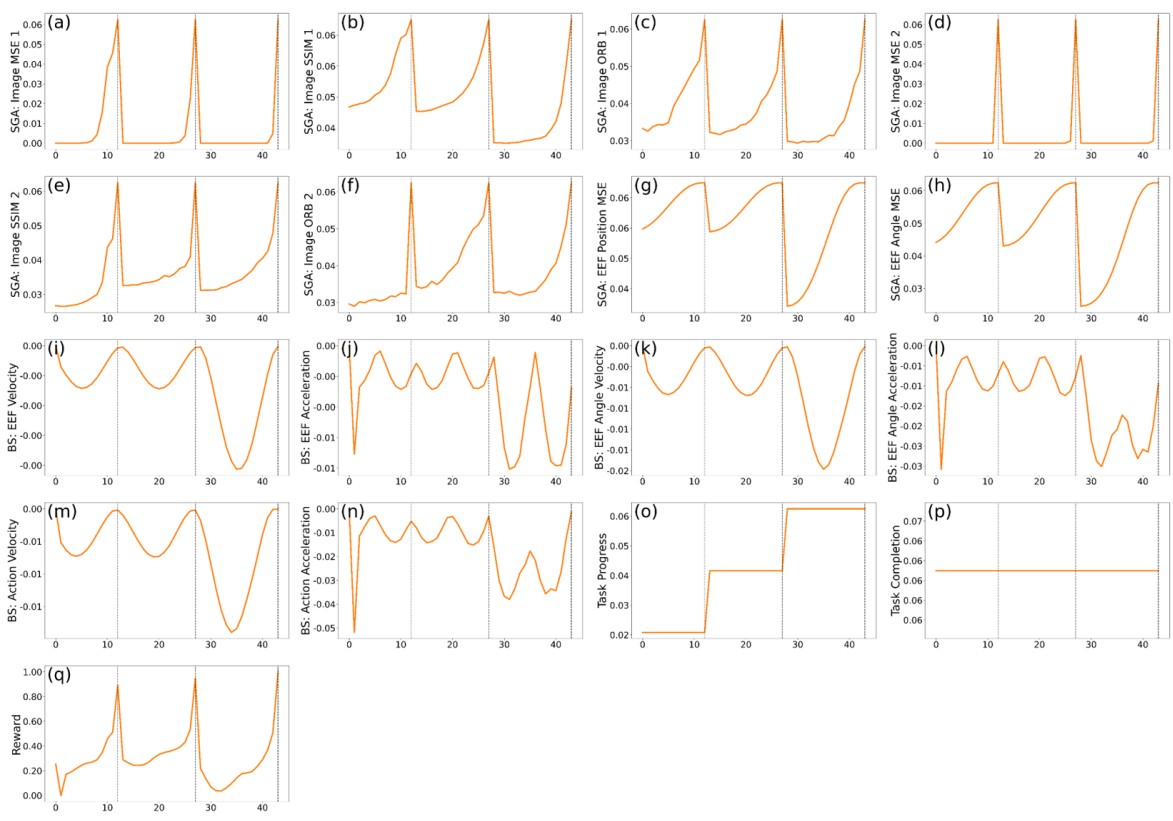

*Figure 13.* The dense reward and reward component of long-horizon tasks with language instructions of "*Pick up the green cup for me*" in the real-world UR5 successful training data.

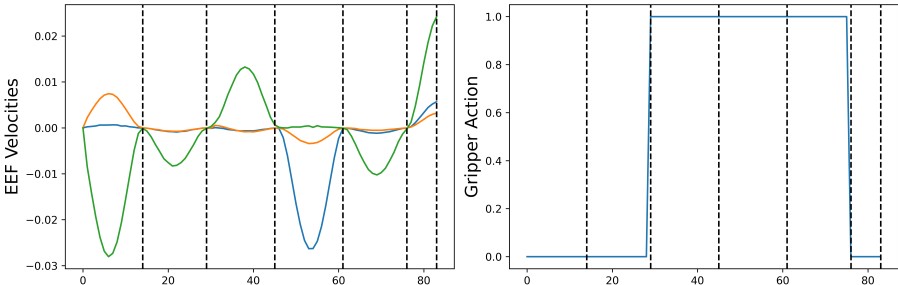

*Figure 14.* The sub-goal division of long-horizon tasks with language instructions of "*Put the smaller blue bowl into the red bowl*" in the real-world UR5 successful training data.

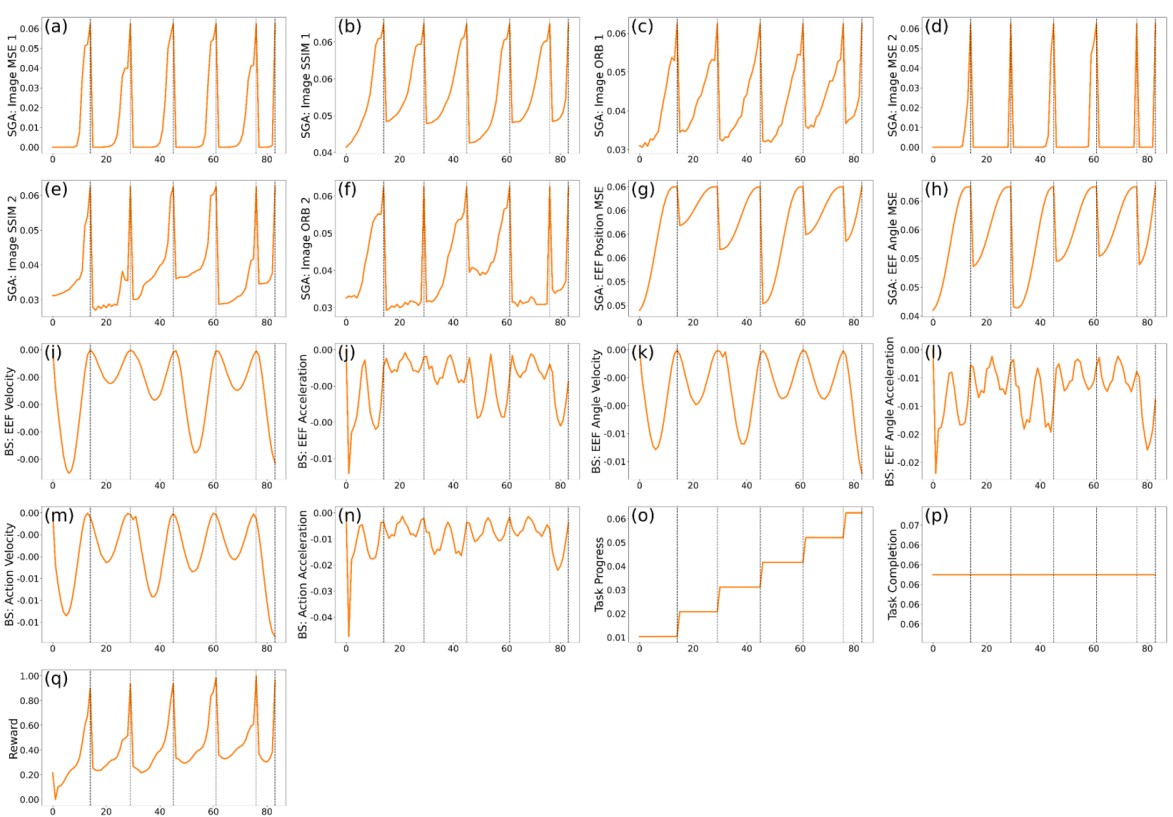

*Figure 15.* The dense reward and reward component of long-horizon tasks with language instructions of "*Put the smaller blue bowl into the red bowl*" in the real-world UR5 successful training data.

(a) Language Instruction: *pick up the red cup for me*

(b) Language Instruction: *put the smaller blue bowl into the red bowl*

(c) Language Instruction: *take the tiger out of the red bowl and put it in the green bowl*

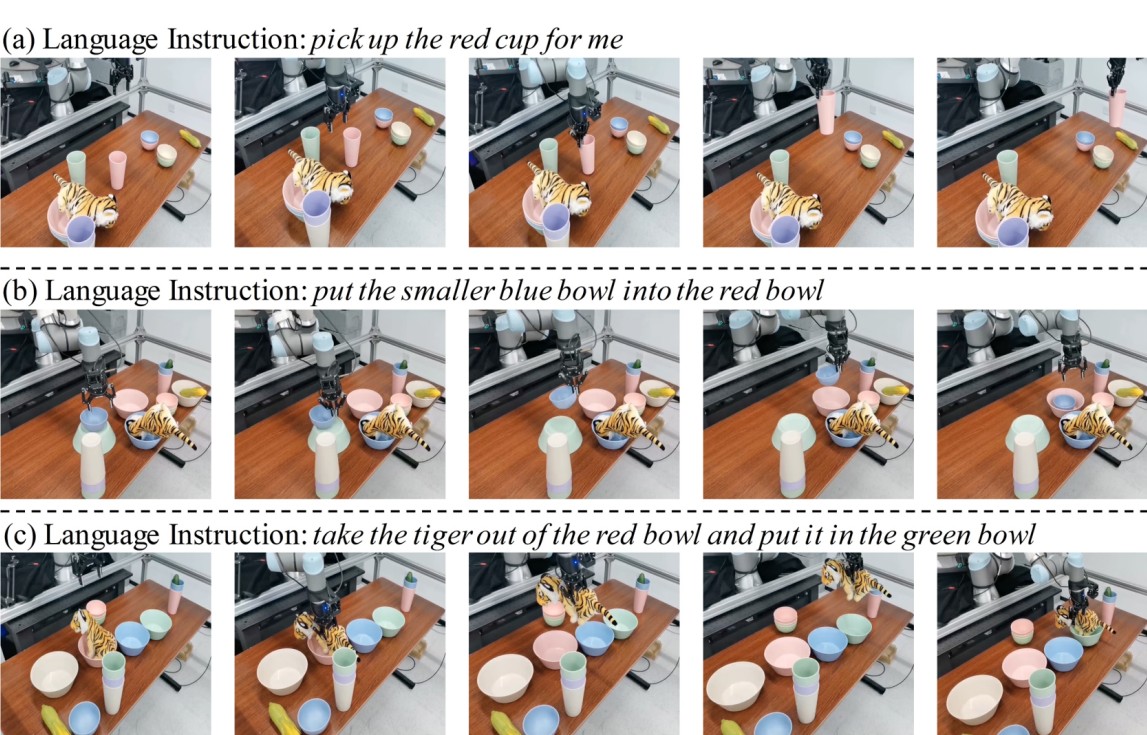

*Figure 16.* Real machine deployment of the proposed ReinboT model. ReinboT can effectively complete real-world pick-and-place tasks with few-shot learning.

