# OpenReview forum: "ReinboT: Amplifying Robot Visual-Language Manipulation with Reinforcement Learning"
_ICML.cc/2025/Conference — ICML 2025 poster_

### Official Review · Reviewer_bfnn · 2025-02-27

**Overall Recommendation:** 3

**Summary:**

The paper introduces ReinboT, a VLA model integrated with offline RL for robotic manipulation. It improves decision-making by predicting dense rewards, which guide the robot to maximize long-term returns. ReinboT outperforms baseline models in simulated and real-world tasks, demonstrating superior few-shot learning and generalization.

## update after rebuttal
I confirm my score. Authors addressed comments and added clarity and results to the original submission.

**Claims And Evidence:**

Yes

**Essential References Not Discussed:**

To the best of my knowledge, no

**Experimental Designs Or Analyses:**

Yes

**Methods And Evaluation Criteria:**

Yes

**Other Comments Or Suggestions:**

- Do experiments in a more complicative environment to test the robustness of ReinboT.
- Do an ablation study on the reward $r_1$ and test it to see whether ReinboT needs the complicative reward design.
- Provide the readers with experience on how to set hyperparameters $\lambda$ and $m$ when adopting ReinboT to other scenarios.

**Other Strengths And Weaknesses:**

Strengths:
- ReinboT’s dense reward mechanism and Return-to-Go (RTG) prediction provide a novel approach to improving long-horizon decision-making in robotic RL. These methods offer creative solutions to sparse reward problems.

Weaknesses:
- The model’s performance is highly sensitive to the choice of $\lambda$ and $m$. If $m$ is too large, the model may become overly optimistic in estimating rewards, leading to poor adaptation and decreased performance. This sensitivity limits ReinboT’s generalization across tasks and datasets.
- While the proposed dense reward function works well for certain tasks, it may not be flexible enough for more complex environments. A more refined reward design may be needed to better capture the nuances of complex tasks.

**Questions For Authors:**

- Why does ReinboT need to predict future image?
- The paper mentioned that "in the classic RL algorithm, maximizing the Q-value is utilized to achieve the best policy model. This implies that applying RL in VLA necessitates the introduction of an additional RL loss function. Such an addition may pose obstacles to the learning process of models like transformers.". Why applying RL in VLA may bring obstacles to transformers? Did you fine-tune VLA with PPO? These results may support the claim more convincingly.

**Relation To Broader Scientific Literature:**

Dense reward design, return-to-go prediction.

**Theoretical Claims:**

The theoretical foundation of ReinboT is not formally established through rigorous proofs or mathematical derivations.

---

> ### Author Rebuttal · Authors · 2025-03-31
>
> We sincerely thank the reviewer for your positive and insightful response to our work. In response to the reviewer's concerns about more baseline comparisons and reward ablation, we have added more experiments. We respond to all concerns point by point.
>
> **Q1**: While the proposed dense reward function works well for certain tasks, it may not be flexible enough for more complex environments.
>
> **R1**: One of the core contributions of our work is to propose a sparse reward densification method for general visual language manipulation tasks. We believe that the design principles of these four rewards will greatly inspire the reward design for more complex environments. More refined and flexible reward design for complex environments is an important future work.
>
> **Q2**: Do experiments in a more complicative environment to test the robustness of ReinboT.
>
> **R2**: We compare ReinboT, OpenVLA and Pi-0 on more challenging real-world tasks. Please see our response (**R1**) to **Reviewer TPwh**.
>
> **Q3**: Do an ablation study on the reward $r_1$ and test it to see whether ReinboT needs the complicative reward design.
>
> **R3**: The ablation experiment of reward $r_1$ is shown in the table below. The experimental results show that the loss of any part of $r_1$ will lead to a decrease in model performance. Therefore, we do need to explore the characteristics of multi-modal manipulation tasks from various aspects of modal information to better characterize the data quality distribution.
> ### Ablation Study on $r_1$
> | Design Choices                 | 1    | 2    | 3    | 4    | 5    | Avg. Length        |
> |--------------------------------|:----:|:----:|:----:|:----:|:----:|:------------------:|
> | **ReinboT (Ours)**             | **0.79** | **0.58** | **0.40** | **0.28** | **0.21** | **2.26**          |
> | W/o sub-goal achievement $r_1$    | 0.72 | 0.50 | 0.32 | 0.20 | 0.12 | 1.87 (-17.2%)     |
> | W/o proprioception state MSE   | 0.72 | 0.52 | 0.33 | 0.24 | 0.20 | 2.01 (-11.1%)     |
> | W/o image state MSE            | 0.75 | 0.50 | 0.33 | 0.27 | 0.22 | 2.07 (-8.4%)      |
> | W/o image state SSIM           | 0.71 | 0.53 | 0.31 | 0.19 | 0.18 | 1.92 (-15.4%)     |
> | W/o image state ORB            | 0.76 | 0.51 | 0.32 | 0.23 | 0.16 | 1.97 (-12.8%)     |
>
> **Q4**: Provide the readers with experience on how to set  $\lambda$ and $m$.
>
> **R4**: The original imitation learning loss is still the main supervisory signal for model learning, while the ReturnToGo prediction is more of an auxiliary role. Therefore, the $\lambda$ needs to be appropriately set smaller, and we recommend starting from around 0.001.
> On the other hand, the $m$ is utilized to control the model’s sensitivity to different expectation levels, thereby adjusting the model’s fitting characteristics for the ReturnToGo distribution. After extensive ablation experiments, we recommend setting the $m$ from around 0.9.
>
> **Q5**: Why does ReinboT need to predict future image?
>
> **R5**: Predicting future images is just consistent with the baseline method (GR-1). The core contribution of ReinboT lies in the introduced reward densification method and the organic integration of the RL return maximization principle in VLA.
>
> **Q6**: Why applying RL in VLA may bring obstacles to transformers? Did you fine-tune VLA with PPO? These results may support the claim more convincingly.
>
> **R6**: Transformer architectures, especially large pre-trained VLMs, are sensitive to noisy gradients introduced by online RL. Directly fine-tuning the entire VLA model using RL loss can destabilize training. Unlike language tasks, robotics online RL requires exploration, which introduces high-variance updates that can corrupt the carefully tuned parameters of the VLM or even cause it to change catastrophically. Although we do not currently fine-tune VLA directly with PPO, recent research [1] confirms the above. Intuitively, in the online actor-critic RL learning setting, the exploration behavior of the policy is prone to produce some bad trajectory states, and the critic network that has just been trained will have difficulty accurately evaluating the value of these trajectory states, thus misleading the fine-tuning of the VLA policy model and causing performance degradation. In the context of multi-modal manipulation tasks, how to effectively achieve accurate estimation of the critic network and reduce training and memory overhead is also an open question. In contrast, our currently proposed ReinboT bypasses the thorny critic network estimation problem and effectively achieves the benefits of maximizing rewards in RL under the offline learning setting. Further extending ReinboT to the online setting is an important future task.
>
> [1] Guo, Y., Zhang, J., Chen, X., Ji, X., Wang, Y. J., Hu, Y., & Chen, J. (2025). Improving Vision-Language-Action Model with Online Reinforcement Learning. arXiv preprint arXiv:2501.16664.

---

> > ### Comment · Reviewer_bfnn · 2025-04-03
> >
> > Thank you for addressing my and other reviewers' comments.

---

> > > ### Author Response · Authors · 2025-04-03
> > >
> > > We are pleased to receive your prompt response and are happy to see that your concern has been fully addressed.
> > >
> > > Thank you again for your insightful and thoughtful comments.

---

### Official Review · Reviewer_TPwh · 2025-03-09

**Overall Recommendation:** 2

**Summary:**

The paper "ReinboT: Amplifying Robot Visual-Language Manipulation with Reinforcement Learning" introduces an end-to-end Vision-Language-Action (VLA) model, ReinboT, which integrates reinforcement learning (RL) principles to improve robotic manipulation. The core idea is to incorporate a dense reward structure to address the issue of mixed-quality training data that limits the performance of existing VLA models based purely on imitation learning. The paper proposes a novel dense reward design that captures various aspects of robotic manipulation, such as sub-goal achievement, task progress, behavior smoothness, and task completion. Additionally, the model leverages a ReturnToGo (RTG) prediction mechanism to guide decision-making by estimating future cumulative rewards. Experimental results show that ReinboT outperforms state-of-the-art models on the CALVIN benchmark and demonstrates superior few-shot learning and generalization to out-of-distribution (OOD) scenarios in real-world robotic tasks.

**Claims And Evidence:**

- **ReinboT improves robotic manipulation accuracy over existing VLA models.** This claim is supported by experimental results showing ReinboT’s superior performance on CALVIN mixed-quality data.

- **Dense reward modeling enhances learning efficiency.** The ablation studies confirm that each component of the dense reward contributes to performance improvements.

- **ReturnToGo prediction enables better decision-making.** The paper presents empirical evidence that RTG-based predictions enhance long-horizon planning.

- **ReinboT generalizes better to unseen tasks.** The few-shot learning and OOD experiments support this claim.

**Essential References Not Discussed:**

N/A

**Experimental Designs Or Analyses:**

The experimental design is comprehensive and well-executed. The paper includes:

- Baseline comparisons with state-of-the-art VLA and RL-integrated models.

- Ablation studies to examine the impact of dense reward components.

- Real world experiments

However, I wonder how Reinbot performs compared to OpenVLA [Kim et al., 2024], Pi-0 [Octo Model Team et al., 2024], or imitation learning + data curation work [Nasiriany et al., 2022; Du et al., 2023; Kuhar et al., 2023].

[Kim et al., 2024] OpenVLA: An Open-Source Vision-Language-Action Model, arXiv 2024

[Octo Model Team et al., 2024] Octo: An Open-Source Generalist Robot Policy,

[Nasiriany et al., 2022] Learning and retrieval from prior data for skillbased imitation learning. CoRL 2022

[Du et al., 2023] Behavior retrieval: Few-shot imitation learning by querying unlabeled datasets. arXiv 2023

[Kuhar et al., 2023] Learning to discern: Imitating heterogeneous human demonstrations with preference and representation learning. CoRL 2023

**Methods And Evaluation Criteria:**

The methodology appears well-structured and appropriate for the problem domain.

**Other Comments Or Suggestions:**

N/A

**Other Strengths And Weaknesses:**

Strengths
- The integration of RL principles into a VLA model is novel and well-motivated.

- The dense reward formulation is carefully designed and empirically validated.

- The experimental results are strong, demonstrating both simulation and real-world efficacy.

Weaknesses and Questions

- A direct comparison with Open-VLA and Pi-0 would provide further clarity on ReinboT’s advantages.

- How does ReinboT compare in terms of inference speed relative to prior VLA models like OpenVLA [Kim et al., 2024], particularly in real-time applications?

- How does Reinbot perform compared to imitation learning + data curation [Nasiriany et al., 2022; Du et al., 2023; Kuhar et al., 2023]?

- the reward engineering seems to require some insights from humans, is there any idea to automate the reward engineering?

[Kim et al., 2024] OpenVLA: An Open-Source Vision-Language-Action Model, arXiv 2024

[Octo Model Team et al., 2024] Octo: An Open-Source Generalist Robot Policy,

[Nasiriany et al., 2022] Learning and retrieval from prior data for skillbased imitation learning. CoRL 2022

[Du et al., 2023] Behavior retrieval: Few-shot imitation learning by querying unlabeled datasets. arXiv 2023

[Kuhar et al., 2023] Learning to discern: Imitating heterogeneous human demonstrations with preference and representation learning. CoRL 2023

**Questions For Authors:**

N/A

**Relation To Broader Scientific Literature:**

The paper builds upon previous works in:

- Imitation learning for VLA models (e.g., GR-1)

- Offline RL and return maximization techniques (e.g., Decision Transformer, Reinformer)

- Reward modeling for robotic control

ReinboT integrates reinforcement learning (RL) principles to improve robotic manipulation.

**Theoretical Claims:**

The paper does not heavily focus on formal theoretical contributions.

---

> ### Author Rebuttal · Authors · 2025-03-31
>
> We sincerely thank the reviewer for your insightful comments, which can further enhance the completeness and thoroughness of our work. In response to the reviewer's concerns about more baseline comparisons, we have added more experiments. We respond to all concerns point by point.
>
> **Q1**: A direct comparison with Open-VLA and Pi-0.
>
> **R1**: We further fine-tuned OpenVLA and Pi-0 on more than 900 real-world ur5 robot arm grasping and placing trajectories (including small square wooden blocks, plastic cups, corn, green peppers and other objects). The training configuration is 8xA100 GPUs, batch size is 8 (for OpenVLA) and 32 (for Pi-0), 40000 steps of fine-tuning. The test results show that our proposed ReinboT has competitive performance with Pi-0, with a success rate of about 0.8, which is better than OpenVLA (Only about 0.35). Compared with ReinboT, Pi-0 is pre-trained on a wider and more diverse range of manipulation data and has more model parameters (Pi-0 model parameters are about 3B, while ReinboT has about 300M). Therefore, the capabilities of ReinboT deserve further development, and expanding its training data volume and network parameter scale is an important future work. On the other hand, we speculate that the main reason for the low success rate of OpenVLA is that it only uses a single image observation and a discretized action space. Local perception makes it difficult to grasp the spatial position of objects, and discretized actions also have serious error accumulation problems in long-horizon manipulation tasks, resulting in poor performance.
>
> **Q2**: How does ReinboT compare in terms of inference speed relative to prior VLA models like OpenVLA?
>
> **R2**: In our ur5 real machine deployment, the inference speeds of ReinboT, Pi-0 and OpenVLA models are about 0.09s, 0.12s and 0.2s respectively. In fact, ReinboT also uses action chunking technology to generate 64 future actions at a time on the ur5 manipulation task. If we execute multiple steps of action (currently only one step of action is executed by default), the inference efficiency of the model can be further improved. In addition, we did not make any additional designs like Pi-0. The ReinboT model is deployed on the local platform, and the OplenVLA and Pi-0 models are deployed on the remote server. Local deployment platform: CPU: 12th Gen Intel(R) Core(TM) i7-12700F; GPU: NVIDIA GeForce RTX 2060/PCIe/SSE2. Remote server platform: CPU: Intel(R) Xeon(R) Platinum 8358 CPU @ 2.60GHz; GPU: NVIDIA A100-SXM4-80GB.
>
> **Q3**: How does Reinbot perform compared to imitation learning + data curation?
>
> **R3**: The core idea of combining imitation learning with data curation is to use a metric that measures the distance between source and target data to filter out high-quality data for imitation learning. Therefore, in our VLA problem setting, an intuitive data curation baseline approach is to use ReturnToGo as a trajectory quality metric and filter out high ReturnToGo data for imitation learning. The performance comparison of the proposed ReinboT and the data curation baseline method on CALVIN is shown in the following table. Experimental results show that ReinboT can achieve superior performance. Compared with data management methods that directly discard low-quality data, ReinboT can have a more fine-grained understanding of the distribution of mixed-quality data, thereby better guiding action generation.
> ### Comparison on CALVIN Mixed-quality Data (Test D)
> | Algorithms                     | 1    | 2    | 3    | 4    | 5    | Avg. Length       |
> |--------------------------------|:----:|:----:|:----:|:----:|:----:|:-----------------:|
> | Imitation learning + data curation | 0.69 | 0.47 | 0.28 | 0.22 | 0.14 | 1.78             |
> | **ReinboT (Ours)**             | **0.79** | **0.58** | **0.40** | **0.28** | **0.21** | **2.26 (+27.0%)** |
>
> **Q4**: the reward engineering seems to require some insights from humans, is there any idea to automate the reward engineering?
>
> **R4**: Previous work used off-the-shelf multi-modal LLM to automatically label robot trajectories with rewards. However, the prompts of LLM still require time and effort to design. The hallucination problem of LLM also significantly affects the accuracy of the designed rewards. In contrast, the rule-based densified reward we proposed is based on the characteristics of long-horizon multi-modal manipulation tasks. It is naturally suitable for manipulation tasks, thus having a wider range of applicability. For new manipulation tasks, we only need to slightly adjust the velocity threshold as the key point criterion and the reward weight used to balance the reward value. This process does not impose an undue burden on researchers' time and energy.
>
> We believe that the reviewer's concerns have been fully addressed. Please convey any additional concerns that hinder the reviewer from improving the score in a timely manner. Thank you again for the reviewer's detailed and thorough review comments.

---

### Official Review · Reviewer_rsGY · 2025-03-14

**Overall Recommendation:** 3

**Summary:**

This paper proposed an RL framework for finetuning the Vision-Language-Action model called ReinBoT, including automatic sub-goal division and reward densification, which is critical to applying RL in VLA training. The authors also use ReturnToGo to replace vanilla challenging value estimation. ReinBoT achieves SoTA performance in CALVIN benchmark.

**Claims And Evidence:**

1. Sub-goal division and reward densification are proved to be useful in Table.1.
2. ReturnToGo estimation is proved to be critical in ablation studies in Table.2.
3. ReinBoT did achieve SoTA performance in CALVIN as demonstrated in Table.1.

**Essential References Not Discussed:**

None.

**Experimental Designs Or Analyses:**

It would be better if they include more than one benchmark.

**Methods And Evaluation Criteria:**

Methods:
The subgoal division is a rule-based specific design for the manipulation arms, which would restrain the generality of this method to other robotic multi-period tasks. The reward densification including subgoal achievement, task progress, and behavior smoothness is effective and general to many other tasks. Dense reward is important for reducing the difficulty for RL agents. The author also employ ReturnToGo estimation directly which alleviates the value estimation problem.

Experiments:
ReinBoT reported a SoTA performance in the CALVIN benchmark. I am not an experienced researcher in VLA area but wanna know why not include more benchmarks like as described in OpenVLA? I think this would much strengthen this paper.

**Other Comments Or Suggestions:**

None.

**Other Strengths And Weaknesses:**

None.

**Questions For Authors:**

1. For the behavior smoothness reward defined in Eq.(6), it seems to encourage to increase the velocity and acceleration. Why this design leads to smooth actions? Or did you miss a minus sign？
2. What are the weights of the four reward components?

**Relation To Broader Scientific Literature:**

ReinBoT could be a future mainstream RL framework in RL for VLA training.

**Theoretical Claims:**

This paper doesn't include theoretical analysis.

---

> ### Author Rebuttal · Authors · 2025-03-31
>
> We sincerely thank the reviewer for your positive response and affirmation of our work. We respond to the reviewer's concerns point by point.
>
> **Q1**: Experiments: ReinBoT reported a SoTA performance in the CALVIN benchmark. I am not an experienced researcher in VLA area but wanna know why not include more benchmarks like as described in OpenVLA? I think this would much strengthen this paper.
>
> **R1**: In the field of VLA for general-purpose tasks, model training generally requires a large amount of training data and computing resources, as well as a hardware platform adapted for real-machine deployment. Therefore, conducting experimental comparisons on a wider range of benchmarks is a time-consuming and laborious task that is not affordable for general laboratories.
> Moreover, in order to provide a direct comparison with previous work (OpenVLA and Pi-0), we further fine-tuned OpenVLA and Pi-0 on more than 900 real-world ur5 robot arm grasping and placing trajectories (including small square wooden blocks, plastic cups, corn, green peppers and other objects). The training configuration is 8xA100 GPUs, batch size is 8 (for OpenVLA) and 32 (for Pi-0), 40000 steps of fine-tuning. The test results show that our proposed ReinboT has competitive performance with Pi-0, with a success rate of about 0.8, which is better than OpenVLA (Only about 0.35). Compared with ReinboT, Pi-0 is pre-trained on a wider and more diverse range of manipulation data and has more model parameters (Pi-0 model parameters are about 3B, while ReinboT has about 300M). Therefore, the capabilities of ReinboT deserve further development, and expanding its training data volume and network parameter scale is an important future work. On the other hand, we speculate that the main reason for the low success rate of OpenVLA is that it only uses a single image observation and a discretized action space. Local perception makes it difficult to grasp the spatial position of objects, and discretized actions also have serious error accumulation problems in long-horizon manipulation tasks, resulting in poor performance.
>
> **Q2**: For the behavior smoothness reward defined in Eq.(6), it seems to encourage to increase the velocity and acceleration. Why this design leads to smooth actions? Or did you miss a minus sign？
>
> **R2**: The reward weight $w_3$ corresponding to the reward $r_3$ in formula (6) is a negative number. Therefore, the overall reward is to suppress the velocity and acceleration of the robot manipulation, thereby enhancing the smoothness of the action.
>
> **Q3**: What are the weights of the four reward components?
>
> **R3**: The four weights are $\frac{0.1}{T_{traj.}}$, $\frac{0.1}{T_{traj.}}$, $\frac{-0.01}{T_{traj.}}$ and $\frac{0.1}{T_{traj.}}$, where $T_{traj.}$ represents the corresponding trajectory length. The reward weights are summarized in Appendix Table 4.

---

> > ### Comment · Reviewer_rsGY · 2025-04-02
> >
> > Thanks for your clarifications!
> >
> > I understand the resource limits. I still recommend aligning symbols in reward weights and add the minus to Eq.6 for better understanding.
> >
> > I prefer to hold my score now.

---

> > > ### Author Response · Authors · 2025-04-03
> > >
> > > We really appreciate your prompt response.
> > >
> > > Regarding the symbol alignment in the reward weights, we will carefully review and adjust the symbols to ensure consistency and clarity throughout the manuscript. This will definitely improve the readability and understanding of our work.
> > >
> > > As for adding the minus sign in Equation 6, we agree that this will make the equation more intuitive and easier to understand. We will modify Equation 6 accordingly and double-check the relevant content for consistency.

---

### Official Review · Reviewer_TPRg · 2025-03-16

**Overall Recommendation:** 4

**Summary:**

This paper introduces ReinboT, a model that integrates reinforcement learning (RL) concepts, specifically the idea of maximizing cumulative dense rewards, into a vision-language-action (VLA) framework for robotic manipulation. The authors propose a reward design that breaks down long-horizon tasks into several sub-goals and factors in elements such as sub-goal achievement, task progress, behavior smoothness, and task completion. By predicting ReturnToGo (RTG) – effectively a measure of future cumulative reward – ReinboT adjusts its policy to maximize these returns within a single end-to-end learning setup. The model is built on a GPT-style transformer backbone that encodes various modalities, including images, language instructions, and proprioceptive data. In experiments on both simulated (CALVIN) and real-world robotic tasks, ReinboT achieves higher success rates and better generalization capabilities than baselines that rely solely on imitation learning or more conventional RL approaches.

**Claims And Evidence:**

The main claims are that (1) incorporating dense reward information into a vision-language-action transformer allows the model to better recognize and leverage mixed-quality offline data, (2) predicting a “maximized” ReturnToGo guides the policy toward actions that yield benefits for long-horizon tasks. The evidence supporting these claims are extensive on the CALVIN benchmark; in particular, the paper does a good job demonstrating that ReinBoT is an effective approach for incorporating elements of RL into a large VLA architecture.

**Essential References Not Discussed:**

I think all important related works are discussed in the paper.

**Experimental Designs Or Analyses:**

The experiments contain large-scale validation on the CALVIN dataset, real-world deployment, as well as various ablation studies. The experimental methodology seems sound and standard in the literature.

**Methods And Evaluation Criteria:**

The paper focuses on success rate and generalization capabilities of ReinBoT. These evaluation criteria are appropriate for the problem: chaining instructions tests long-horizon robustness, and OOD scenarios reveal how well the model extrapolates from its training distribution. The design of dense rewards is argued to reflect real-world aspects of manipulation tasks (like smoothness and sub-goal achievement), which is a practical choice for multi-stage robotics tasks. The combination of these evaluation metrics provides a solid overview of how well the method performs in both standard and challenging conditions.

**Other Comments Or Suggestions:**

N/A.

**Other Strengths And Weaknesses:**

Weaknesses:
1. I think the criteria for keypoints in demonstration can be sensitive to the quality of demonstrations; for instance, when demonstrations have mis grasps, there will be false positive keypoints. Some discussions on the sensitivity of quality of demonstrations are warranted for more effective real-world usage.
2. Comparison against other VLA architectures and RL framework, such as OpenVLA with actor-critic, can help demonstrate the particular combination of the proposed approach.

**Questions For Authors:**

My questions and suggestions are stated above.

**Relation To Broader Scientific Literature:**

The paper’s main contribution to the broader field is showing that a single end-to-end architecture can incorporate an RL optimization perspective (via expectile regression) without needing a separate Q-function or an online adaptation stage can be an effective way to build large-scale VLAs. While prior works on VLA are discussed, more in-depth experimental validation can be helpful.

**Theoretical Claims:**

This paper does not introduce new theoretical claims.

---

> ### Author Rebuttal · Authors · 2025-03-31
>
> We sincerely thank the reviewer for your high affirmation and recognition of our work. We respond to the reviewer’s concerns point by point.
>
> **Q1**: I think the criteria for keypoints in demonstration can be sensitive to the quality of demonstrations; for instance, when demonstrations have mis grasps, there will be false positive keypoints. Some discussions on the sensitivity of quality of demonstrations are warranted for more effective real-world usage.
>
> **R1**: The designed rule-based densification reward contains four items: sub-goal achievement, task progress, behavior smoothness, and task completion. The task completion reward is result-oriented and is used to determine whether the entire trajectory is successful. The other rewards are process-oriented and are used to assist the robot's decision-making actions. In some cases, false positive keypoints appear when there are incorrect grasping behaviors in the demonstration. However, the behavior smoothness reward suppresses the velocity and acceleration of the robot's body state and actions, thus penalizing such incorrect behaviors of violent back-and-forth grasping. Therefore, the final densified reward can be resistant to such noise in the demonstration data to a certain extent and is not so sensitive to the quality of such demonstrations.
>
> **Q2**: Comparison against other VLA architectures and RL framework, such as OpenVLA with actor-critic, can help demonstrate the particular combination of the proposed approach.
>
> **R2**: It is currently an open problem to efficiently combine VLA models such as OpenVLA with the actor-critic framework in RL. This is because OpenVLA discretizes the action space, while the standard actor-critic algorithm models in a continuous action space. Moreover, in the context of multi-modal manipulation tasks, it is also a problem to efficiently achieve accurate estimation of the critic network and reduce training and memory overhead. In contrast, the proposed ReinboT circumvents the thorny critic network estimation problem and effectively implements the benefit of maximizing the return in RL.

---

> > ### Comment · Reviewer_TPRg · 2025-04-06
> >
> > Thank you for your response -- I will maintain my original acceptance score.

---

> > > ### Author Response · Authors · 2025-04-07
> > >
> > > We are pleased to see your reply and your concern has been fully addressed.
> > >
> > > Thank you again for your high recognition and affirmation of our work.

---

### Decision · Program_Chairs · 2025-05-01

**Decision:**

Accept (poster)

**Comment:**

The paper proposes ReinboT, a framework for finetuning vision-language-action (VLA) models using techniques from reinforcement learning (RL). In an effort to address the challenges of imitation learning from mixed-quality demonstrations, ReinboT provides dense reward feedback to VLA models by decomposing long-horizon tasks into subgoal sequences that are used together with behavior smoothness and task completion to define a dense reward signal. ReinboT employs ReturnToGo (RTG), a predicted measure of the future cumulative reward, to further support the fine-tuning of VLAs. Evaluations on the CALVIN benchmark demonstrate that ReinboT outperforms contemporary baselines, while an ablation study reveals the contributions of the key components of the framework.

The paper was reviewed by four referees, who agree on a number of the paper's strengths and weaknesses. Among them, at least three of the reviewers find that the paper provides clear evidence regarding the benefits of the ways in which ReinboT uses RL methods to improve VLA training, including the advantages of subgoal decomposition and the use of the ReturnToGo objective, with ReinboT achieving state-of-the-art performance. The reviewers raised a few concerns with the paper as initially submitted, notably the importance of including comparisons to other VLA architectures and RL frameworks (e.g., OpenVLA with actor-critic). In their rebuttal, the authors noted that the incorporation of actor-critic methods is challenging since the actions are discrete, though it is worth noting the existence of work adapting actor-critic methods to discrete action spaces [Petros, 2019]. During the rebuttal, the addition of a comparison to OpenVLA and Pi-0 suggests that ReinboT is competitive, however more a more extensive evaluation is required before any conclusions can be drawn. That said, the paper provides interesting insights into the ways in which techniques from RL can be utilized to improve the training of VLAs, which would be of interest to many in the robot learning community.

The AC notes that TPwh, who provided the most critical review, did not engage in the discussion process, despite several attempts on the part of the AC.


Christodoulou, Petros. "Soft actor-critic for discrete action settings." arXiv preprint arXiv:1910.07207 (2019).